# The Effect of 45S5 Bioglass and Ag, Cu, or Zn Addition on the Crystal Structure, Properties, and Antibacterial Effect of Bulk Ti23Zr25Nb Biocomposites

**M. Marczewski** [1] , **M. Jurczyk** [1,]*, **P. Pecyna** [2] , **M. Ratajczak** [2] , **M. Gajecka** [2,3] and **M. U. Jurczyk** [4]

[1]  Institute of Materials Science and Engineering, Poznan University of Technology, Jana Pawla II 24, 61-138 Poznan, Poland; mateusz.p.marczewski@doctorate.put.poznan.pl

[2]  Faculty of Pharmacy, Department of Genetics and Pharmaceutical Microbiology, Poznan University of Medical Sciences, Swiecickiego 4, 60-781 Poznan, Poland; pa.pecyna@gmail.com (P.P.); ratajczak.magdalena@wp.pl (M.R.); gamar@man.poznan.pl (M.G.)

[3]  Institute of Human Genetics, Polish Academy of Sciences, Strzeszynska 32, 60-479 Poznan, Poland

[4]  Division of Mother's and Child's Health, Poznan University of Medical Sciences, Polna 33, 60-535 Poznan, Poland; mjur@poczta.onet.pl

*  Correspondence: mieczyslaw.jurczyk@put.poznan.pl; Tel.: +48-61-665-3508

**Abstract:** In the present study, the crystal structure, microstructure, mechanical, corrosion properties, and wettability of bulk Ti23Zr25Nb-x45S5 Bioglass (x = 0, 3, 6, 9 wt.%) and Ti23Zr25Nb—9 wt.% 45S5 Bioglass composites with the addition of 1 wt.% Ag, Cu, or Zn were synthesized and their properties studied. The hardness of these biomaterials is at least two times higher and the elastic modulus lower in comparison to commercial purity (CP) microcrystalline $\alpha$-Ti. The mechanically alloyed Ti23Zr25Nb—9 wt.% 45S5 Bioglass composite was more corrosion resistant in Ringer's solution than the bulk Ti23Zr25Nb alloy. Surface wettability measurements revealed the higher surface hydrophilicity of the bulk synthesized composites. The antibacterial activity of Ti23Zr25Nb-based composites containing silver, copper, or zinc against reference strain *Streptococcus mutans* ATCC 25175 was studied. In vitro bacterial adhesion indicated a significantly reduced number of *S. mutans* on the bulk Ti23Zr25Nb-BG-Ag (or Cu, Zn) plate surfaces in comparison to the microcrystalline Ti plate surface. Ultrafine-grained Ti23Zr25Nb-BG-Ag (or Cu, Zn) biomaterials can be considered to be the next generation of dental implants.

**Keywords:** Ti-Zr-Nb alloy; 45S5 Bioglass; mechanical alloying; structural properties; antibacterial activity; *S. mutans*

---

## 1. Introduction

Beta-type titanium alloys are an interesting group of biomaterials. Mechanical alloying is the technology of producing them with much smaller grain sizes than conventional methods such as arc melting. The structure evolution of titanium-based alloys with different beta-stabilizers (Nb, Zr, and Mo) produced with mechanical alloying has been already examined [1,2]. These alloys were described as really interesting in terms of their properties such as the Young modulus. Moreover, further modification leads to their improvement, including better corrosion resistance [3].

Particular interest should be focused on modifying these alloys by the formation of biocomposite materials. That allows us to decrease their Young modulus, increase the corrosion resistance, and antibacterial properties. Biocomposites formed by adding 45S5 Bioglass to these alloys during milling can lead to interesting results such as higher biocompatibility, hardness, and a lower Young modulus.

Moreover, Ag content increases the antibacterial behavior of alloys and the formation of foams improves the osteointegration process [4,5].

Modification of titanium alloys with bioglass and bioglass-reinforced composites was also done with surface engineering methods as electrophoretic deposition (EPD) [6–13], magnetron sputtering [14,15], pulsed electron deposition [16] and plasma spraying [17]. Hydroxyapatite and bioglass co-coatings produced with laser engineering net shaping (LENS) should also be a matter of scientific interest because of their promising corrosion resistance, wear-resistance, and interaction with bone cells [18]. A potential medical application has also been documented with an anodic bonding method of titanium with 52S4.6 and 45S5 Bioglasses [19]. Electrophoretically deposited films also showed good properties on a non-titanium alloy like stainless steel and Nitinols [20]. The nanobioglass composited with silver [21], metal oxides such as $TiO_2$ [22], and graphene oxides [23,24] was done with the use of the sol–gel technique. Depending on the modification technique, the properties of bioglass can be significantly improved in terms of their bioactivity, hemocompatibility, antibacterial activity, and cell proliferation properties.

The Zn addition was proved to have antibacterial properties in the coatings, either as an implanted ion with plasma immersion technology [25] or as a doping element for the calcium phosphates films [26–28]. Zn is also successfully added to the coatings with other antibacterial elements such as Ag [29].

Ti-Cu alloys with high antibacterial properties were also produced. In these alloys, the crucial factor is not only the Cu content but also precipitation of the $Ti_2Cu$ phase with the best effect achieved with the homogenous fine precipitate. The Cu addition also leads to the hardening effect of these alloys. According to most of the research, at least 5 wt% of Cu is needed to have the right antibacterial effect against some of the well-known bacteria: *Escherichia coli*, *Staphylococcus aureus*, *Streptococcus mutans* and *Porphyromonas gingivalis* [30–39]. Copper can also be successfully added to non-titanium alloys such as steels, magnesium alloys, cobalt alloys, and zinc alloys. Its addition not only allows the use of the antibacterial properties of that element but also improves its corrosion resistance and mechanical properties [40].

Ti-Ag alloys are also a matter of great importance for the development of materials with antibacterial ability. As in Ti-Cu alloys, the intermetallic $Ti_2Ag$ phase is crucial and its homogenous and nano-scale precipitation could improve their antibacterial properties [41,42]. Silver containing bioglass nanoparticles (AgBGs) produced with the sol–gel method were also created. This type of particle could also be interesting in terms of bioactivity and the antibacterial improvement in the biomedical field of applications [43]. Producing of antibacterial alloys containing silver is a deeply examined topic in biomedical science. Titanium alloys are not the only modified materials with this additive. One of the newest examples of using silver in material engineering is the antibacterial, biodegradable Fe-Mn-Ag alloy [44].

Surface engineering of implants is relevant for safeguarding them against the formation of bacterial biofilms. The use of Zn, Cu, Ag (ion-implanted, etc.), is the selected way of doing that [45,46].

In this work, the ultrafine-grained Ti23Zr25Nb alloy and its composites with 45S5 Bioglass as well as with Ag, Cu, or Zn were synthesized by the application of mechanical alloying and powder metallurgy. As was mentioned in the previous articles considering the production of beta-type Ti-based alloys this method is seen as the most useful in contrast to the conventional ones. It allows the production of ultra-fine grained beta-type titanium alloys in a wide range of the concentrations and with the use of the elements with significantly different melting points [1,2,47]. Grain refinement can lead to further improvement of properties and increased hardness. Mechanically alloyed materials were also proven to have better biological properties [48,49].

The reason for the modification with 45S5 Bioglass was to improve the properties which would lower the Young modulus and improve the corrosion resistance of the non-modified Ti23Zr25Nb alloy. Its corrosion resistance has been revealed as worse than that of microcrystalline titanium in our previous work [47]. This type of bioglass provides good cell growth and differentiation of

osteoblasts due to the low phosphate content and the widest clinical application among all other bioglasses [50,51]. The addition of well-known elements with antibacterial properties such as Ag, Cu, and Zn was to enhance the antibacterial response of the material, which should make the produced material interesting for use in the biomedical field. The influence of the microstructure and chemical composition of Ti23Zr25Nb and Ti23Zr25Nb-BG-Ag (or Cu, Zn) composites on the crystal structure, microstructure, mechanical properties, corrosion behavior, surface wettability, and antibacterial activity against a reference strain of *Streptococcus mutans* were investigated in detail.

## 2. Materials and Methods

### 2.1. Sample Preparation

Mechanical alloying was done with a SPEX 8000 Mixer Mill (SPEX$^{®}$ Sample Prep, Metuchen, NJ, USA) and round bottom stainless vials. To produce the alloy, the composites vials was filled in the Ar atmosphere inside a glove box Labmaster 130 (MBraun, Garching, Germany) with following precursors: Ti (Alfa Aesar, 99.9% purity, <45 µm average particle size, CAS: 7440-32-6), Nb (Sigma Aldrich, 99.8% purity, <45 µm average particle size, CAS: 7440-03-1), 45S5 Bioglass (Mo-Sci GL0160P; 53 µm average particle size, 45% $SiO_2$, 24.5% $Na_2O$, 24.5 CaO, 6% $P_2O_5$), Ag (Alfa Aesar, 99.9% purity, 4–7 µm particle size, CAS: 7440-22-4) powders, Zn (Sigma Aldrich, 99% purity, 30–100 mesh, CAS: 7440-66-6) granules, Cu (Sigma Aldrich, 99% purity, CAS: 7440-50-8) turnings, and Zr fillings from a sponge (Sigma Aldrich, ≥99%, CAS: 7440-50-8). The experiments were carried out on 6 different Ti23Zr25Nb (at.%)-based composites. 45S5 Bioglass composition was varied from 3 to 9 wt.% (to alloy powder ratio). The composite with 9 wt.% content of 45S5 Bioglass was additionally doped with 1 wt.% (to 45S5 Bioglass composite powder ratio) of 3 antibacterial elements (Ag, Zn, Cu). The content of antimicrobial elements was chosen to be as low as possible to introduce the antibacterial properties with as small an impact possible on the other properties such as its biocompatibility and limitation of the additional ions release. Mechanical alloying for all composites was carried out for 16 h. Mechanically alloyed powders were consolidated with conventional cold compaction and sintering. All of the samples were consolidated with the same process parameters. Cold compaction pressure was equal to 600 MPa. Sintering was conducted in an 800 °C temperature for 30 min (allowing the consolidation of the powders with not too much grain growth) and followed with fast water cooling in a quartz tube filled with Ar. The temperature choice was based on previous research of the Ti-Zr-Nb alloys and also Ti-Mo alloys, which proves the increase in β-phase content in most of these types of materials with increase in temperature. Moreover, the highest possible temperature at which it was possible to achieve a fully single β-phase structure in Ti23Zr25Nb (at.%) alloy was 800 °C, which was chosen as the most suitable [2,47]. Bulk sample dimensions were 6 mm diameter and 4 mm in height. The schematic representation of the mechanism behind the mechanical alloying of biocomposites is presented in Figure 1. Additionally, the summarization of mass ratios of precursors and optimization parameters is presented in Table 1.

**Table 1.** The mass ratios of output precursors and optimization parameters for preparation of the studied Ti23Zr25Nb–based biocomposites (TNZ—Ti23Zr25Nb (at.%) alloy [1,47]; BG—45S5 Bioglass; AB—antibacterial additive; T—milling time; CP—compaction pressure; S—sintering temperature).

| Material | BG/TNZ [%] | AB/TNZ-BG [%] | T [h] | CP [MPa] | S [°C] |
|---|---|---|---|---|---|
| Ti23Zr25Nb-3BG | 3 | - | | | |
| Ti23Zr25Nb-6BG | 6 | - | | | |
| Ti23Zr25Nb-9BG | 9 | - | | | |
| T23Zr25Nb-9B-Ag | 9 | 1 | 16 | 600 | 800 |
| T23Zr25Nb-9B-Cu | 9 | 1 | | | |
| T23Zr25Nb-9B-Zn | 9 | 1 | | | |

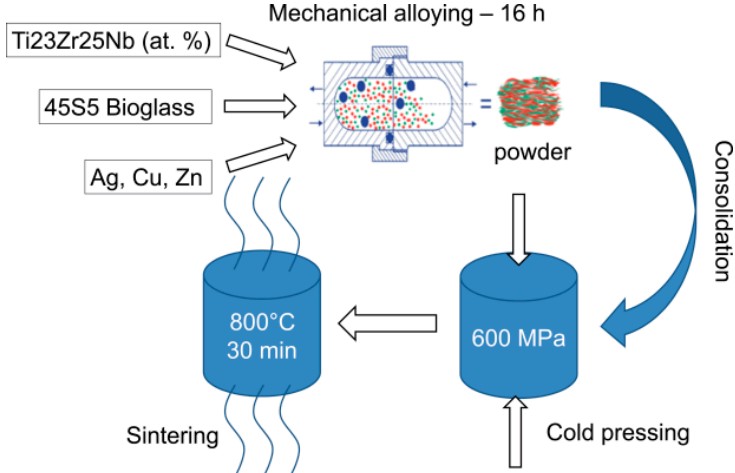

**Figure 1.** Schematics representation of the mechanism behind the mechanical alloying of biocomposites.

## 2.2. Materials Characterization

The crystal structure of produced samples in the powder and bulk stage was evaluated with the Panalytical Empyrean XRD equipment with CuKα radiation (Malvern Panalytical B.V., Almelo, The Netherlands). The microstrain and crystallite size of the powders after 16 h of mechanical alloying was calculated with the uniform deformation model (UDM) of the Williamson–Hall analysis method [52]. Rietveld refinement with Maud software was carried out to estimate the phase amount and lattice parameters of the following phases:

- Ti (α) (ref. code 01-071-4632) hexagonal P63/mmc
- Ti (β) (ref. code 01-074-7075) cubic Im-3m
- $Ti_2ZrO$ (ref. code 01-072-1881) hexagonal P6/mmm
- $Nb_5Si_3P$ (ref. code 00-051-0788) hexagonal P63/mcm

The refinement was done with the use of the Marquardt least-squares algorithm. The following pattering fitting parameters were calculated: $R_{wp}$—weighted pattern residual indicator; $R_{exp}$—expected residual indicator; S—the quality of the fit (lower than 2 for all refinements).

The one selected Ti23Zr25Nb-based (at.%) composite with 9 wt.% 45S5 Bioglass content was examined with a scanning electron microscope (SEM, VEGA 5135 Tescan, Brno, Czech Republic) with an energy dispersive spectrometer (EDS, PTG Prison Avalon, Princeton Gamma Tech., Princeton, NY, USA) calibrated by the typical Cu calibration procedure. The goal of the analysis was to obtain the EDS mapping for componental elements of the material: Ti, Zr, Nb, Si, Ca, Na, P, Ag, Cu, Zn. Additionally, the final powders after the mechanical alloying of Ti23Zr25Nb-BG composites were observed with the same instrument at two different magnifications to present both morphology and particle size. The porosities of the samples were calculated from the histograms in GIMP software based on the micrographs of non-etched samples obtained from the optical microscope Olympus GX51 (Olympus, Tokyo, Japan). Calculations included the pores of different sizes, revealed at two different magnifications.

Mechanical properties of the materials were tested with a Vickers microhardness tester Innovatest Nexus (Innovatest, Maastricht, The Netherlands). The measurement parameters were as follows: an applied load of 300 g and loading time of 10 s. Ten hardness measurements were conducted for each sample. Nanoindentation tests for selected samples with different contents of 45S5 Bioglass were also done with Picodentor HM500 (Fischer Technology Inc., Windsor, CT, USA). The measurement was carried out with DIN 50 359/ISO 14577 standard and load parameters: F = 300 mN/20 s, C = 5 s. Based on load-depth curves, the indentation modulus (EIT) was calculated.

A Kruss-DSA25 digital camera and Kruss-Advanced 1.5 software (Krüss, Hamburg, Germany) was used to measure diiodomethane and glycerol contact angles as well as calculate surface free

energies with dispersive (polar) parts for every bulk sample; each sample polished with the $Al_2O_3$ suspension, flushed with alcohol and dried. The ellipse method was used to fit the drop shape formed with 2 µL of the mentioned fluids. Tests were conducted at ambient conditions (23 °C) and three times for each sample.

Corrware and Corrview software, as well as the Solartron 1285 potentiostat (Solartron Analytical, Farnborough, UK), were used for the potentiodynamic corrosion measurements of all composites polished with 600 grit grinding paper and cleaned with the ethanol inside an ultrasonic bath before each measurement. Open-circuit potential measurement (OCP) was first conducted for 60 min. In the created electrochemical cells, Ringer's solution (NaCl: 9 g/L, KCl: 0.42 g/L, $CaCl_2$: 0.48 g/L, $NaHCO_3$: 0.2 g/L) was the electrolyte and Ag/AgCl was used as the reference electrode. Each material was examined three times with both corrosion potential and current calculated from Tafel curves. The measurement was conducted in the range of −1 V to 2.5 V vs. OCP, and the scan speed was equal to 1 mV/s.

### 2.3. Assessment of Biofilm Formation Inhibition

The strain of *S. mutans* was obtained from the American Type Culture Collection (ATCC 25175). A detailed description of the assessment of biofilm formation inhibition was previously presented [4].

The quantitative dilutions and surface spread method were applied to assess the bacterial adherence, after 4 h and 20 h, respectively, to the experimental biomaterial surfaces. All experiments were repeated three times. Statistical software R version 3.0.1 was applied to determine whether any significant difference existed in bacterial numbers in the antibacterial experiments. Analysis of variance (ANOVA) followed by Tukey's honest significant difference (HSD) test were performed on the bacterial counts. The statistical significance was defined as $p < 0.05$.

### 3. Results

Figure 2 presents the diffractograms of TiZrNb-based composites powders in the function of milling time. The 3 wt.% content of 45S5 Bioglass led to the longer form of the new Ti ($\beta$) phase in comparison to the Ti23Zr25Nb alloy [1]. The Ti ($\beta$) phase appeared after 10 h of milling and 16 h of mechanical alloying was needed to produce the single structure Ti ($\beta$) powder. For the biocomposites with the 6 and 9 wt.% 45S5 Bioglass content, after 10 h hours of milling peaks from all starting phases Ti ($\alpha$), Nb/Ti ($\beta$), and Zr were still visible. However, in the case of the Ti23Zr25Nb-6BG composite after 16 h of milling, only Ti ($\beta$) peaks were visible with trace contents of others. The Ti23Zr25Nb-9BG composite still had a multi-phase structure after 16 h of milling with the presence of other starting phases among Ti ($\beta$). Milling between 10 h and 16 h led only to the further fraction of the structure of the powder because of their plastic deformation and hardening caused by the high density of dislocations and the amorphization of the material [53].

The content of 45S5 Bioglass also decreased the crystallite size of the materials in the as-milled state (Figure 3), which was equal to 2.6 nm for the 9 wt.% 45S5 Bioglass content. Further reduction could also be conducted with the 1 wt.% content of silver and made it equal to 2.4 nm. Both powders of Ti23Zr25Nb-9BG and Ti23Zr25Nb-9BG-Ag only produced samples with negative microstrains which corresponded to the tensile strains. Copper and zinc led to the increase in the crystallite size—8.4 and 11.1 nm, respectively, with positive microstrains corresponding to the compression strains.

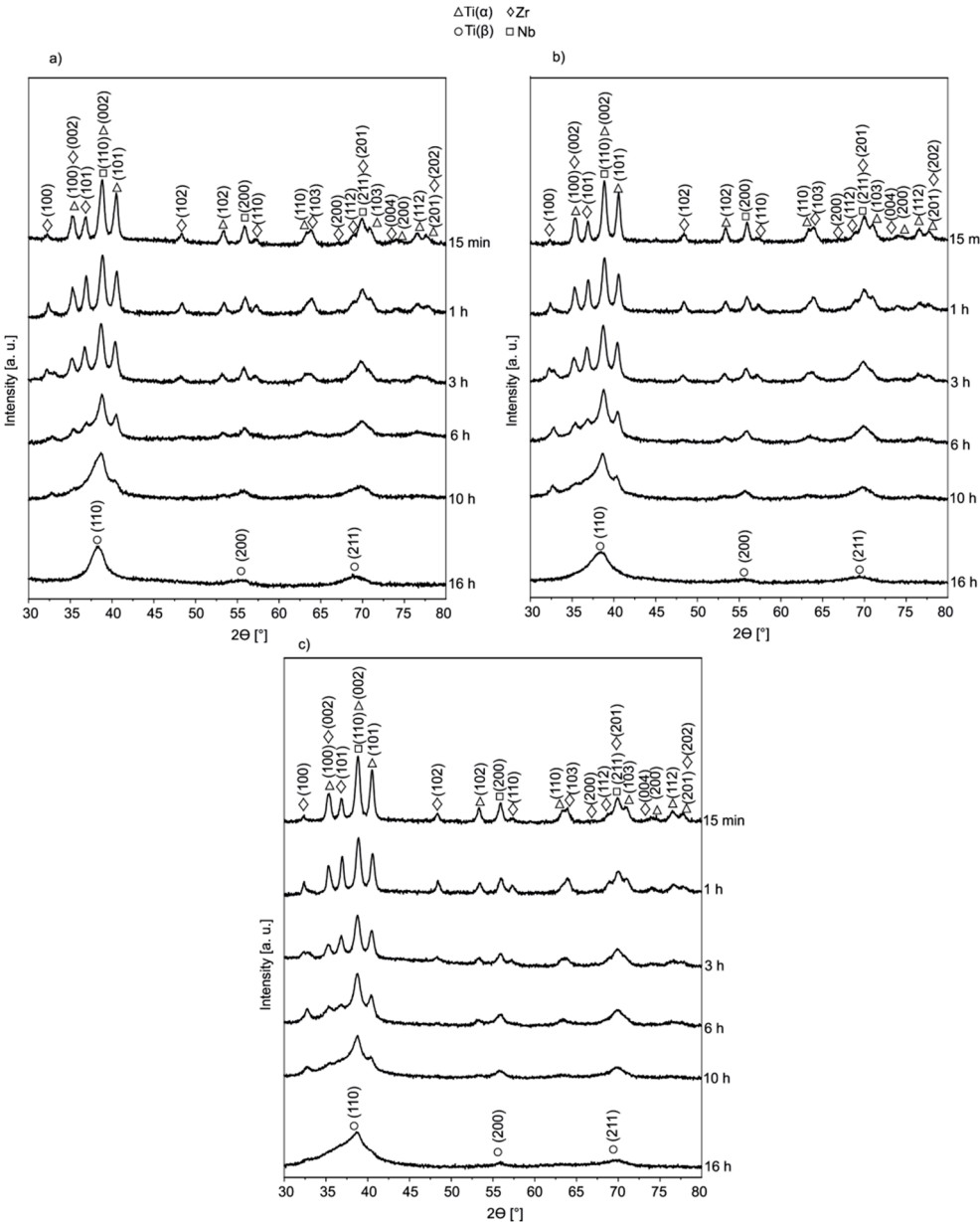

**Figure 2.** XRD spectra of Ti23Zr25Nb-based composites powders with (**a**) 3 wt.%, (**b**) 6 wt.%, and (**c**) 9 wt.% 45S5 Bioglass content mechanically alloyed over different times.

The micrographs of mechanically alloyed (16 h of MA) powders are presented in Figure 4. The morphology of powders particles of all composites was in an irregular shape within the wide size range in contrast to the non-modified Ti23Zr25Nb alloy [47]. Most of them varied from not strongly agglomerated particles in sizes of less than 10 microns to highly agglomerated particles in sizes of hundreds of microns with the biggest particles in sizes of approximately 500 μm. The variation of the particle size was much higher than in the non-modified alloys [1]. The structure of powder was lamellar, with the ceramic phases on the surface of the particles increasing in volume with the bioglass content.

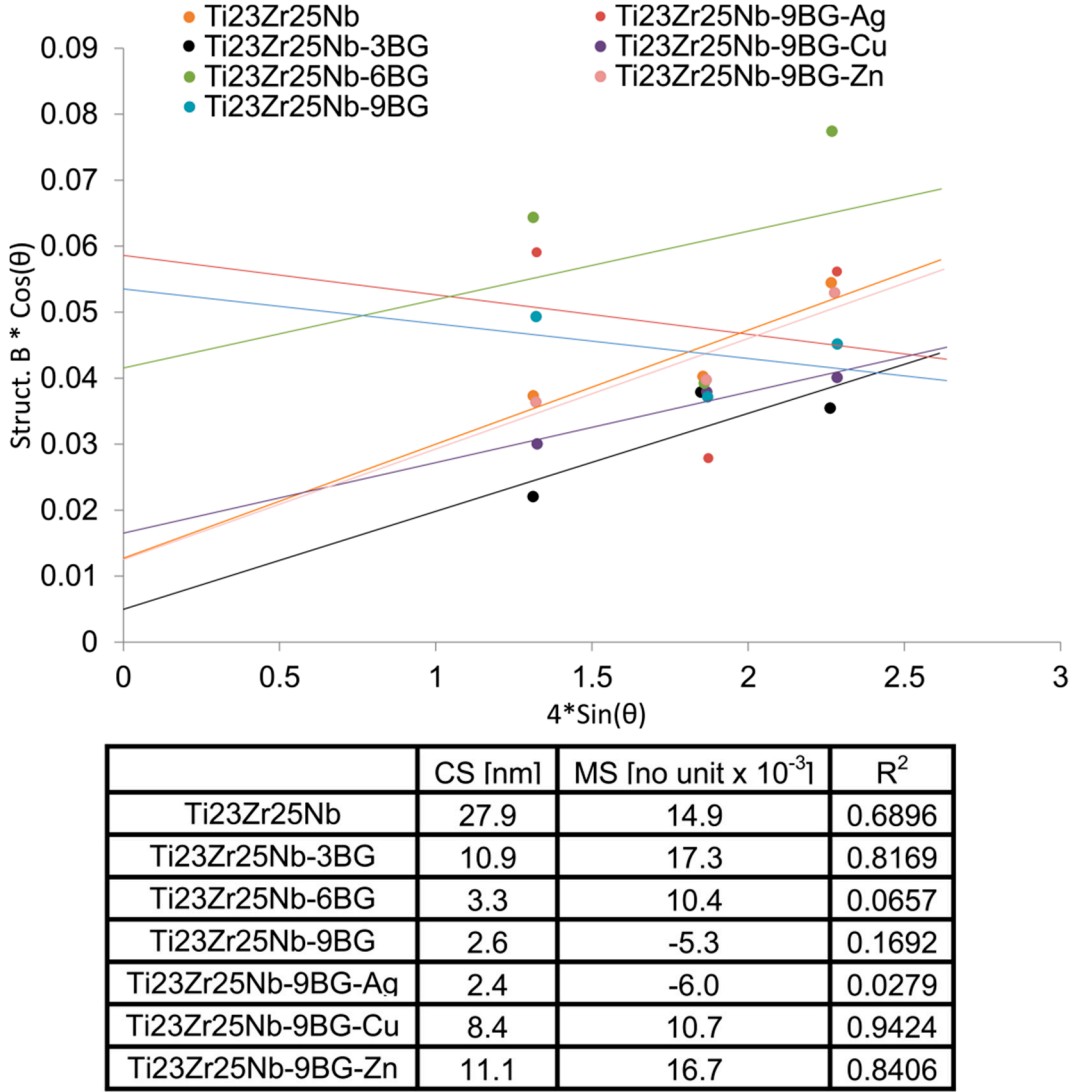

| | CS [nm] | MS [no unit x $10^{-3}$] | $R^2$ |
|---|---|---|---|
| Ti23Zr25Nb | 27.9 | 14.9 | 0.6896 |
| Ti23Zr25Nb-3BG | 10.9 | 17.3 | 0.8169 |
| Ti23Zr25Nb-6BG | 3.3 | 10.4 | 0.0657 |
| Ti23Zr25Nb-9BG | 2.6 | -5.3 | 0.1692 |
| Ti23Zr25Nb-9BG-Ag | 2.4 | -6.0 | 0.0279 |
| Ti23Zr25Nb-9BG-Cu | 8.4 | 10.7 | 0.9424 |
| Ti23Zr25Nb-9BG-Zn | 11.1 | 16.7 | 0.8406 |

**Figure 3.** Linear Williamson–Hall plots with estimated crystallite size (CS) and microstrain (MS) factors based on the XRD spectra of Ti23Zr25Nb-based composites powders after 16 h of mechanical alloying in contrast to the mechanically alloyed Ti23Zr25Nb alloy for 10 h.

The cold compaction and sintering of composites did not allow the formation of the single-phase Ti ($\beta$) structure (Figures 5 and 6). The amount of Ti ($\beta$) decreased to 65.02% in the Ti23Zr25Nb-based composite with the 9 wt.% 45S5 Bioglass content. The addition of Cu and Zn led to its further reduction below 60 wt.% (Table 2). However, despite not having the single-phase Ti ($\beta$) structure, Ti ($\beta$) was still the dominant phase in all the produced materials. The other phases appearing in materials were Ti ($\alpha$), $Ti_2ZrO$, and $Nb_5Si_3P$. The Ti23Zr25Nb-based composite with the 3 wt.% content of 45S5 Bioglass was still the material with the highest amount of the Ti ($\beta$) phase which was above 95%.

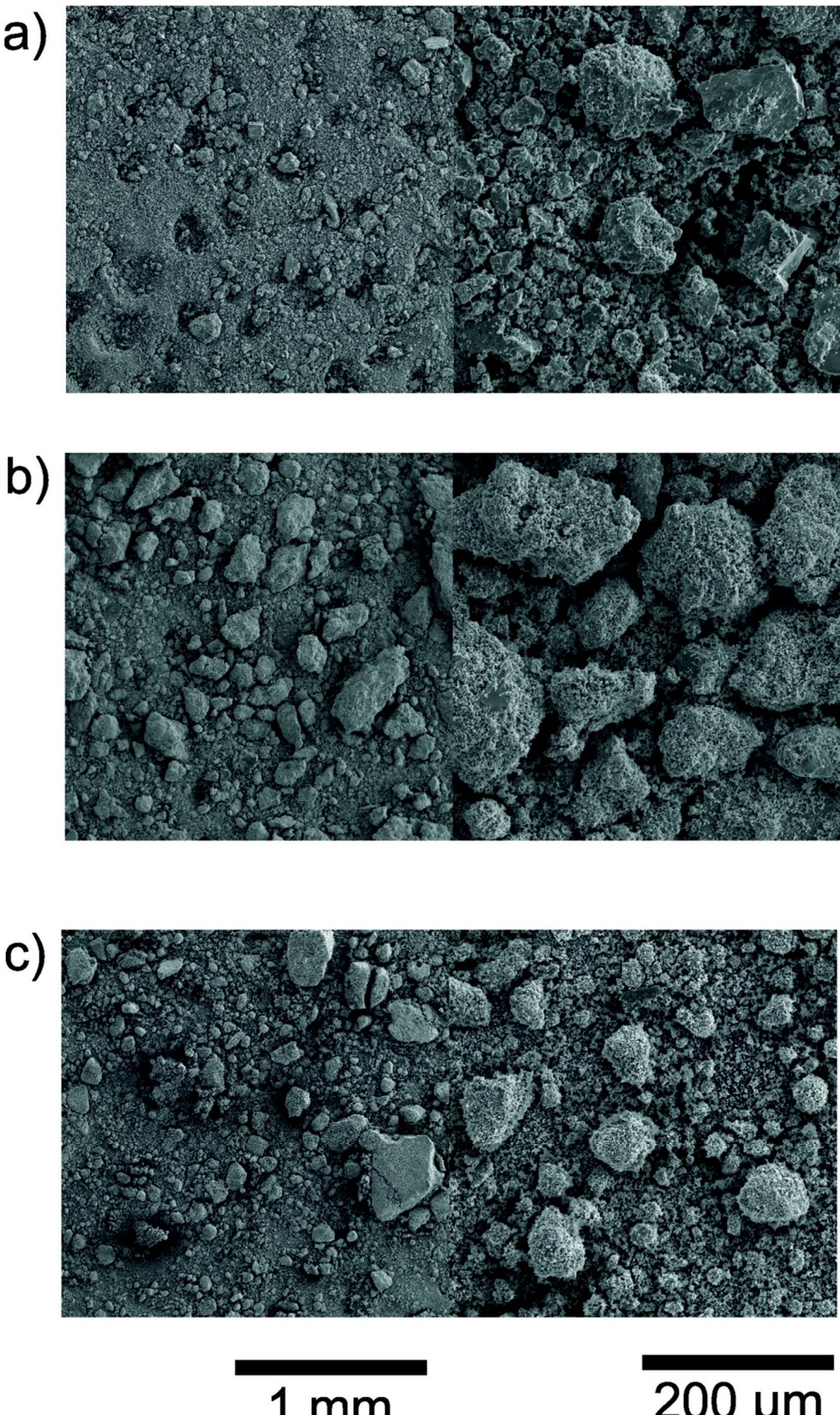

**Figure 4.** SEM microphotographs of Ti23Zr25Nb-based composites powder agglomerates with the (**a**) 3 wt.%, (**b**) 6 wt.%, and (**c**) 9 wt.% 45S5 Bioglass content mechanically alloyed for 16 h.

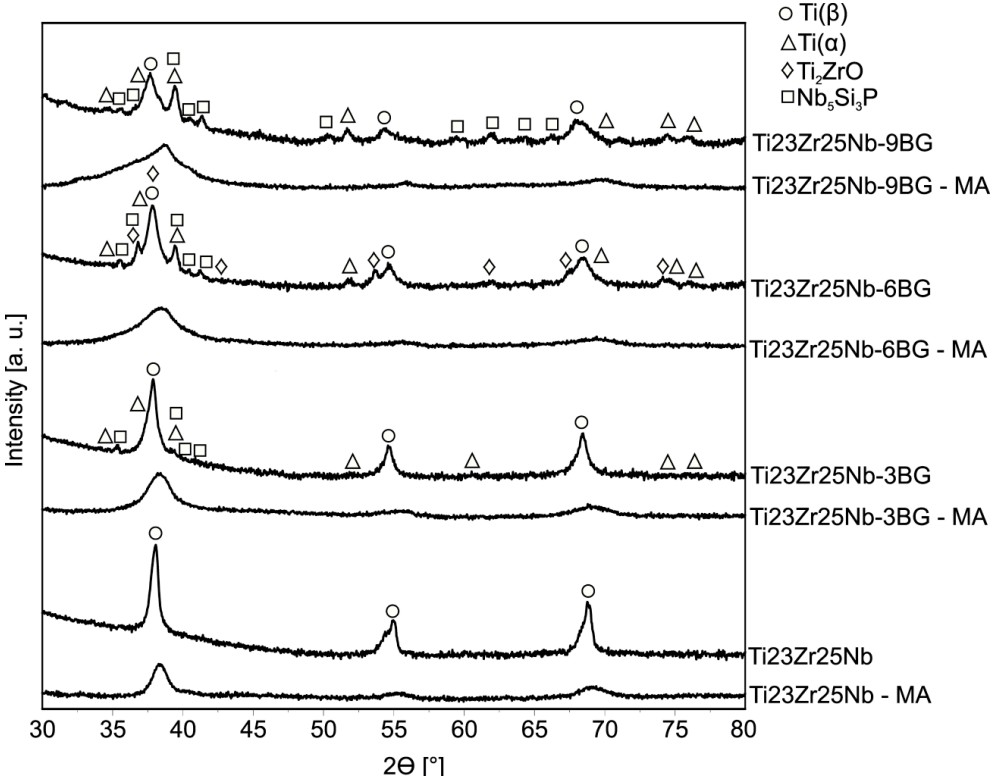

**Figure 5.** XRD spectra of Ti23Zr25Nb-based composites with the 3, 6, and 9 wt.% 45S5 Bioglass content sintered at 800 °C for 0.5 h in an argon atmosphere in contrast with the mechanically alloyed powders milled for 16 h and the Ti23Zr25Nb alloy sintered with the same parameters and mechanically alloyed for 10 h.

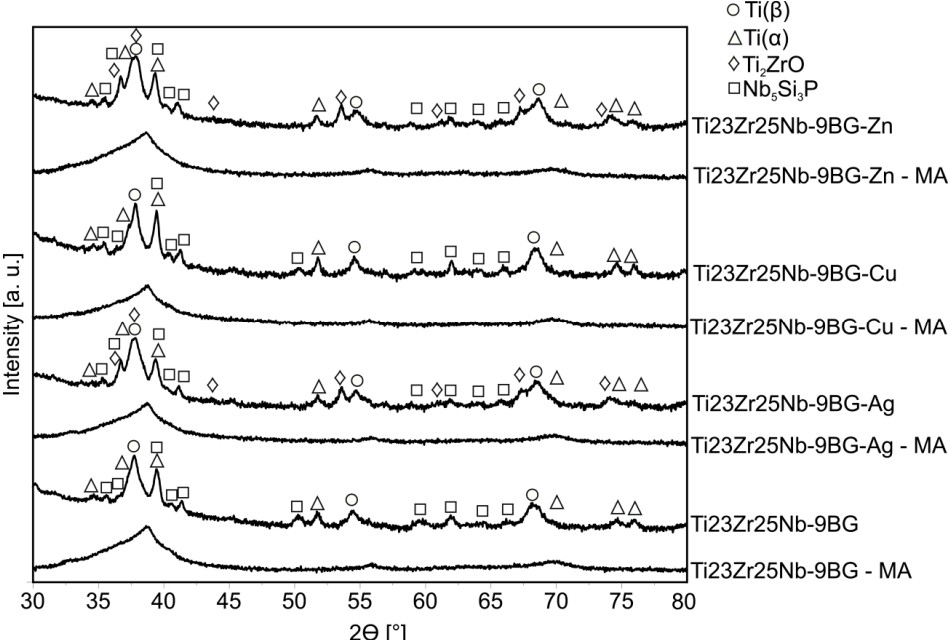

**Figure 6.** XRD spectra of Ti23Zr25Nb-based composites with the 9 wt.% 45S5 Bioglass and 1 wt.% Ag, Cu, Zn content sintered at 800 °C for 0.5 h in an argon atmosphere in contrast with mechanically alloyed powders for milled for 16 h and the based Ti23Zr25Nb-9BG composite sintered with the same parameters and mechanically alloyed for 16 h.

**Table 2.** Crystallographic data analysis of bulk Ti23Zr25Nb-based composites with the 3, 6, 9 wt.% 45S5 Bioglass and 1 wt.%. Ag, Cu, Zn content in contrast to the Ti23Zr25Nb alloy.

| Sample | Tiα | | | | Tiβ | | | Ti$_2$ZrO | | | | Nb$_5$Si$_3$P | | | | Rwp [%] | Rexp [%] | S |
|---|---|---|---|---|---|---|---|---|---|---|---|---|---|---|---|---|---|---|
| | a [Å] | c [Å] | V [Å3] | PA [%] | a [Å] | V [Å3] | PA [%] | a [Å] | c [Å] | V [Å3] | PA [%] | a [Å] | c [Å] | V [Å3] | PA [%] | | | |
| Ti23Zr25Nb | - | - | - | - | 3.3524 (2) | 37.68 (1) | 100 | - | - | - | - | - | - | - | - | 6.28 | 3.65 | 1.72 |
| Ti23Zr25Nb 3BG | 2.9987 (35) | 4.8192 (136) | 37.53 (20) | 2.08 | 3.3602 (2) | 37.94 (1) | 96.38 | - | - | - | - | 7.7633 (145) | 5.3173 (135) | 277.53 (1.74) | 1.54 | 4.65 | 3.4 | 1.37 |
| Ti23Zr25Nb 6BG | 2.9967 (9) | 4.7992 (35) | 37.32 (5) | 9.83 | 3.3577 (2) | 37.86 (1) | 75.63 | 4.7683 (11) | 3.0281 (7) | 59.62 (4) | 10.47 | 7.7296 (34) | 5.3123 (27) | 274.87 (38) | 4.07 | 3.83 | 3.22 | 1.19 |
| Ti23Zr25Nb 9BG | 3.0003 (6) | 4.8085 (21) | 37.49 (3) | 28.13 | 3.3720 (4) | 38.34 (2) | 65.02 | - | - | - | - | 7.7132 (25) | 5.2967 (20) | 272.9 (28) | 6.86 | 4.52 | 3.44 | 1.31 |
| Ti23Zr25Nb 9BG-Ag | 3.0011 (8) | 4.7970 (32) | 37.42 (5) | 15.3 | 3.3579 (4) | 37.86 (2) | 65.98 | 4.7858 (10) | 3.0329 (8) | 60.16 (4) | 13.73 | 7.7575 (29) | 5.3213 (22) | 277.33 (32) | 4.99 | 3.97 | 3.1 | 1.28 |
| Ti23Zr25Nb 9BG-Cu | 2.9946 (4) | 4.8229 (12) | 37.46 (2) | 31.89 | 3.3629 (3) | 38.03 (1) | 59.1 | - | - | - | - | 7.7398 (27) | 5.3117 (21) | 275.57 (30) | 9.00 | 5.02 | 3.38 | 1.49 |
| Ti23Zr25Nb 9BG-Zn | 3.0033 (5) | 4.8188 (21) | 37.64 (3) | 17.13 | 3.3557 (3) | 37.79 (1) | 54.68 | 4.7897 (7) | 3.0301 (6) | 60.20 (3) | 19.47 | 7.7671 (24) | 5.3331 (18) | 278.63 (26) | 8.73 | 3.64 | 2.96 | 1.23 |

EDS mapping confirms the homogenous distribution of all componential elements of the composites on the micro-scale (Figure 7). There were no inhomogeneities in the bulk materials which could have appeared during the consolidation of mechanically alloyed powders. Moreover, maps in Figure 8 presents the distribution of all antibacterial activities (Ag, Cu, and Zn) in the biocomposite matrix.

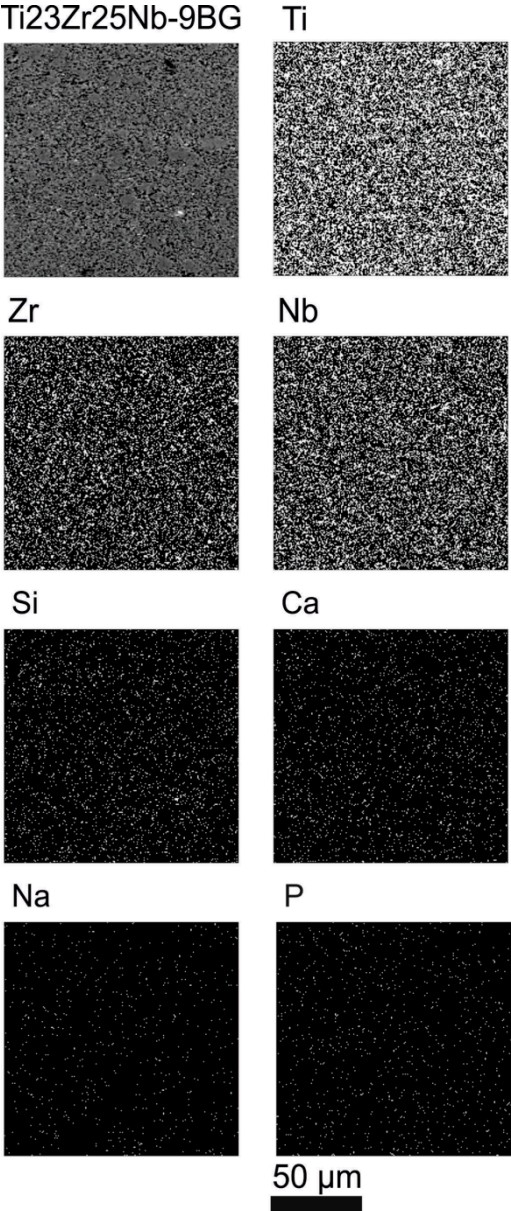

**Figure 7.** Energy-Dispersive Spectroscopy (EDS) maps of Ti, Nb, Zr, Si, Ca, Na in the Ti23Zr25Nb-based composite with the 9 wt.% 45S5 Bioglass content and within the area of the scan micrograph.

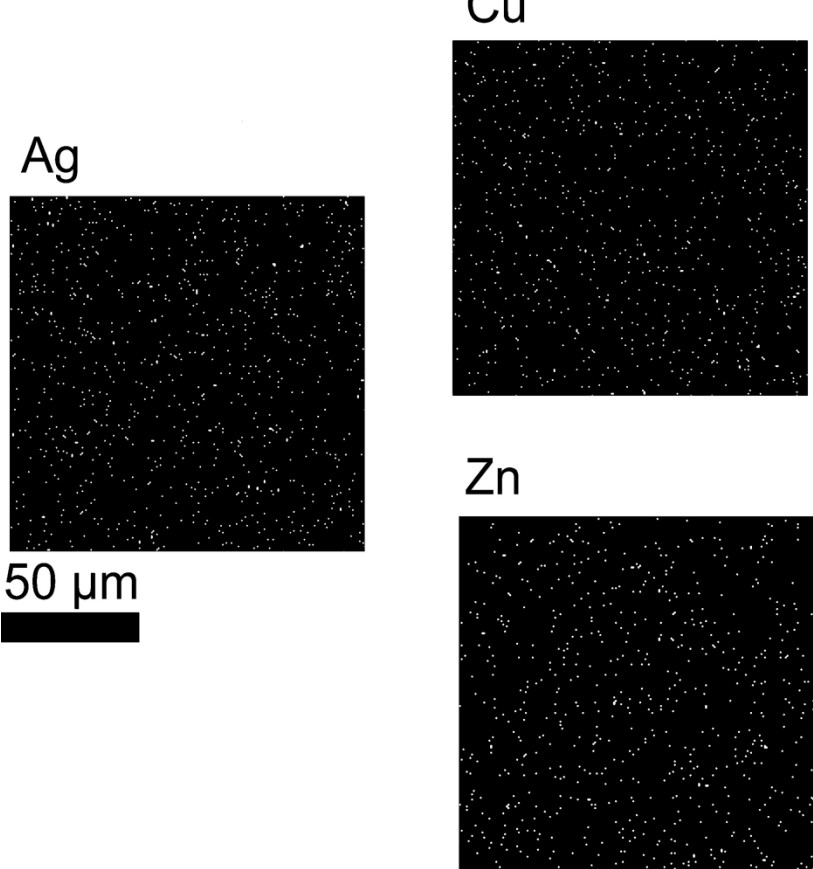

**Figure 8.** Energy-Dispersive Spectroscopy (EDS) maps of Ag, Cu and Zn in the bulk Ti23Zr25Nb-based composite with the 9 wt.% 45S5 Bioglass and 1 wt.% Ag, Cu, Zn.

The porosities of the biocomposites were calculated and are presented in Table 3 and differ between 13.7% (Ti23Zr25Nb-9BG-Zn) and 22.8% (Ti23Zr25Nb-9BG). The morphology of the pores is additionally presented in the micrographs in Figure 9. Porosities were homogeneously distributed on the microscale. Additionally, in Ti23Zr25Nb-6BG and slightly in Ti23Zr25Nb-9BG, a large number of porosities in the size of 10–100 μm were additionally visible with the lower magnifications.

**Table 3.** Porosities calculated with the use of optical micrographs and histograms in GIMP software.

| Material | Porosity [%] |
|---|---|
| Ti23Zr25Nb-3BG | 15.4 ± 3.6 |
| Ti23Zr25Nb-6BG | 22.1 ± 8.9 |
| Ti23Zr25Nb-9BG | 22.8 ± 6.5 |
| Ti23Zr25Nb-9BG-Ag | 14.0 ± 4.1 |
| Ti23Zr25Nb-9BG-Cu | 18.3 ± 7.8 |
| Ti23Zr25Nb-9BG-Zn | 13.7 ± 4.6 |

The mechanical properties of produced composites were superior to commercially pure titanium in terms of their biomedical application. The Young modulus was significantly lower and was lowest for the Ti23Zr25Nb-based composite with the 9 wt.% 45S5 Bioglass content in comparison to the 140.9 ± 2.0 GPa of titanium [1]. The Young modulus of this composite was in conjunction with the high porosity (around 20%) and the lowest hardness (218 HV0.3) (Figure 10 and Table 4). Despite the hardness similar to titanium, the drastic reduction of the Young modulus made it close to the modulus of human bone. Contact angles of all the materials were lower than 90 for both diiodomethane and

glycerol. It confirmed the good wettability of the produced composites. Surface free energies (SFEs) were similar and differed from 34.6 mN/m (Ti23Zr25Nb-6BG) to 42.3 mN/m (Ti23Zr25Nb-9BG-Ag).

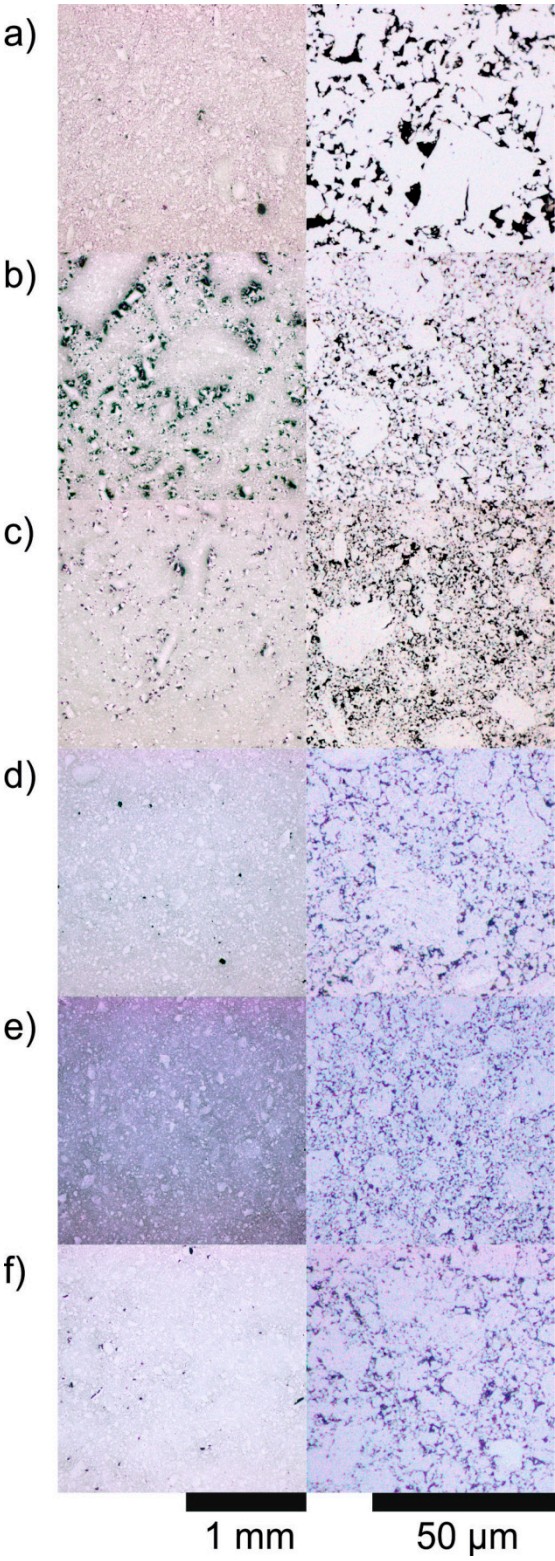

**Figure 9.** Micrographs of non-etched biocomposites revealing porosities (black in the graph) at 2 different magnifications: Ti23Zr25 (at.%) with (**a**) 3 wt.%, (**b**) 6 wt.% and (**c**) 9 wt.% of 45S5 Bioglass content and Ti23Zr25Nb-9BG doped with (**d**) Ag, (**e**) Cu and (**f**) Zn.

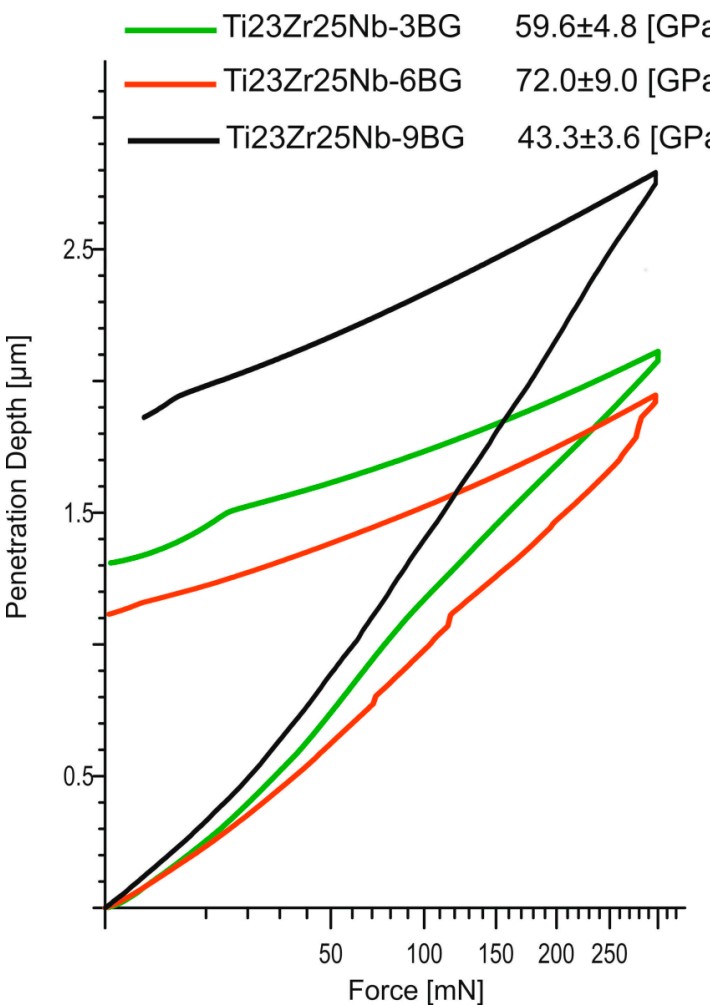

**Figure 10.** Load-depth curves of bulk Ti23Zr25Nb-based composites with the 3, 6, and 9 wt.% 45S5 Bioglass content.

**Table 4.** Vickers hardness (HV0.3), surface free energy (SFE), dispersive and polar parts of SFE, diiodomethane, and glycerol contact angles for Ti23Zr25Nb-based composites with the 3, 6, 9 wt.% 45S5 Bioglass and 1 wt.% Ag, Cu, Zn content in contrast to the Ti23Zr25Nb alloy.

| Sample | HV0.3 | CA [M] Diiodomethane [°] | CA [M] Glycerol [°] | SFE [mN/m] | Disperse [mN/m] | Polar [mN/m] |
|---|---|---|---|---|---|---|
| Ti23Zr25Nb | 375 ± 33 | 62.2 ± 9.0 | 64.6 ± 4.9 | 35.1 ± 10.0 | 27.4 ± 5.6 | 7.7 ± 4.4 |
| Ti23Zr25Nb-3BG | 315 ± 36 | 50.8 ± 7.4 | 70.2 ± 4.5 | 36.1 ± 6.5 | 33.8 ± 4.4 | 2.3 ± 2.1 |
| Ti23Zr25Nb-6BG | 321 ± 22 | 56.0 ± 6.2 | 68.1 ± 7.4 | 34.6 ± 5.7 | 30.9 ± 3.7 | 3.7 ± 2.0 |
| Ti23Zr25Nb-9BG | 218 ± 23 | 57.7 ± 4.4 | 61.5 ± 4.2 | 36.9 ± 3.3 | 29.9 ± 2.5 | 7.0 ± 0.8 |
| Ti23Zr25Nb-9BG-Ag | 322 ± 24 | 44.9 ± 3.8 | 57.5 ± 2.4 | 42.3 ± 4.2 | 37.0 ± 2.2 | 5.3 ± 2.0 |
| Ti23Zr25Nb-9BG-Cu | 329 ± 49 | 49.3 ± 2.4 | 66.7 ± 4.7 | 37.5 ± 2.5 | 34.7 ± 1.3 | 2.8 ± 1.2 |
| Ti23Zr25Nb-9BG-Zn | 387 ± 49 | 50.5 ± 6.6 | 77.8 ± 3.1 | 34.6 ± 4.4 | 33.9 ± 3.9 | 0.7 ± 0.5 |

The corrosion resistance of Ti23Zr25Nb-9BG is also similar to that of titanium and significantly better than that of the Ti23Zr25Nb alloy sintered at the same temperature—800 °C (Figure 11) which was based on the lower corrosion current and higher corrosion potential results. The corrosion current can be further limited by doping this material with silver. However, the corrosion potential of the Ti23Zr25Nb-9BG-Ag composite was higher and there was no clear and visible passivation as in the Ti23Zr25Nb-9BG composite. There was no passivation also in the Ti23Zr25Nb-9BG-Cu alloy (Figure 12).

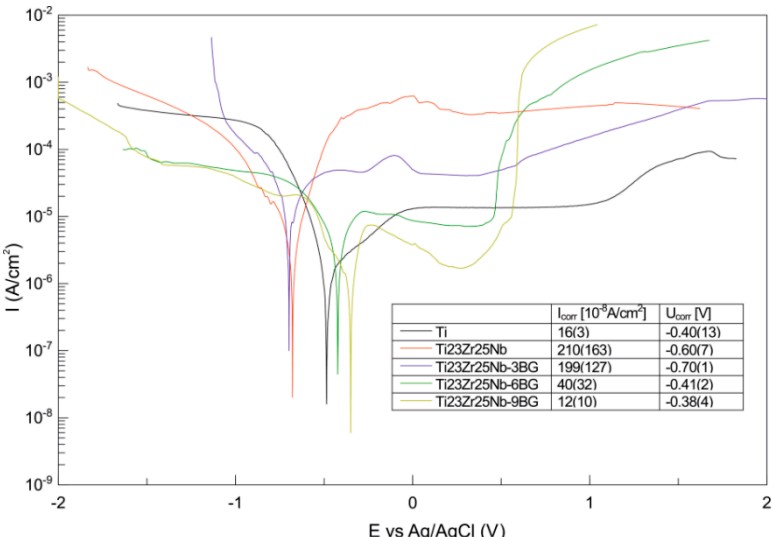

**Figure 11.** Potentiodynamic curves of bulk Ti23Zr25Nb-based composites with the 3, 6, and 9 wt.% 45S5 Bioglass content in contrast to cp-Ti and the Ti23Zr25Nb alloy.

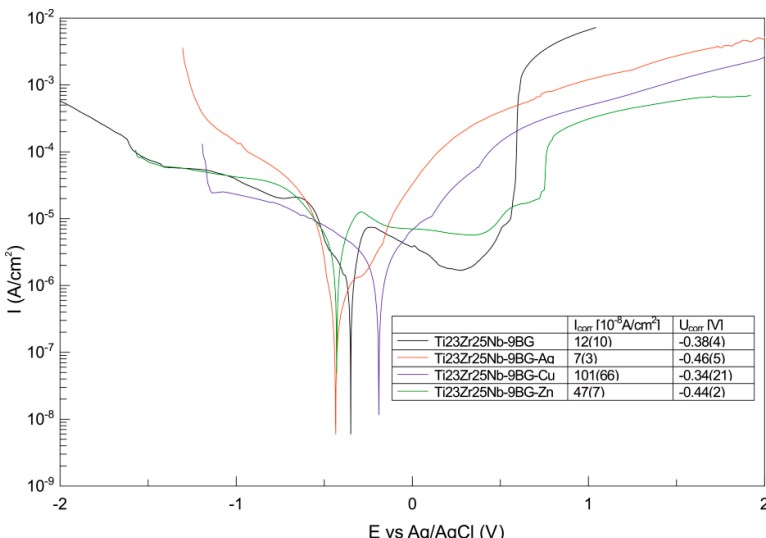

**Figure 12.** Potentiodynamic curves of bulk Ti23Zr25Nb-based composites with the 9 wt.% 45S5 Bioglass and 1 wt.% Ag, Cu, Zn content in contrast with the Ti23Zr25Nb-9BG composite.

Figure 13 shows the results of viable bacteria adhered to the different experimental material surfaces when exposed to *S. mutans* ATCC 25175. The bacterial adhesion was significantly reduced on the surface of Ti23Zr25Nb-9BG-Ag, Ti23Zr25Nb-9BG-Cu, and Ti23Zr25Nb-9BG-Zn composites compared to microcrystalline titanium. These biomaterials were observed to have significantly lower adhesion levels ($p < 0.05$) of *S. mutans* ATCC 25175, suggesting that these composites have inhibited biofilm formation. On the other hand, many bacteria were found on the Ti23Zr25Nb, Ti23Zr25Nb-9BG composites, as shown in Figure 13, displaying that these composites have low antibacterial activity against the reference strain of *S. mutans* (Table 5).

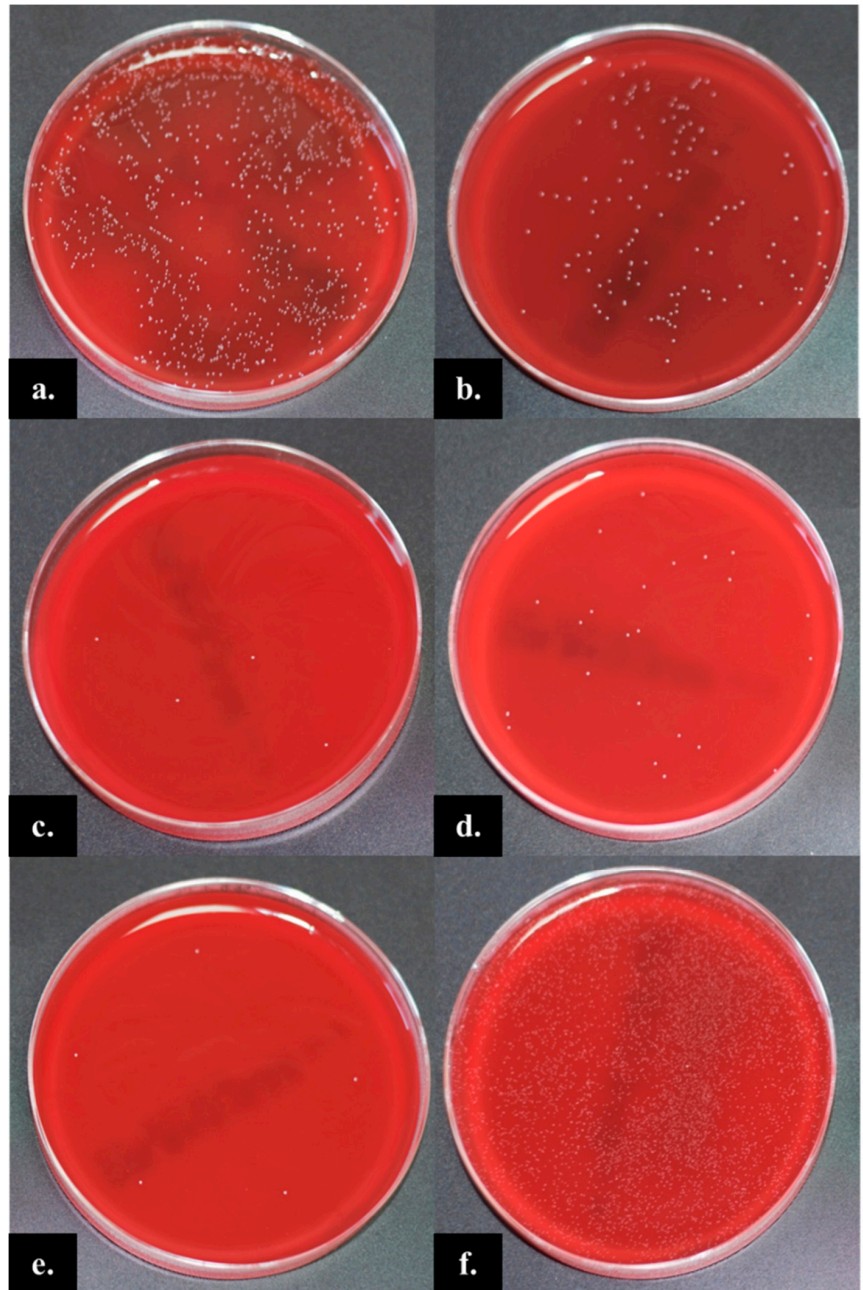

**Figure 13.** Antibacterial activity against *S. mutans* ATCC 25175 growth on agar plates from the same dilution after 24 h incubation on different composites: (**a**) Ti23Zr25Nb, (**b**) Ti23Zr25Nb-9BG, (**c**) Ti23Zr25Nb-9BG-Ag, (**d**) Ti23Zr25Nb-9BG-Cu, (**e**) Ti23Zr25Nb-9BG-Zn, (**f**) control (Ti).

**Table 5.** Antibacterial activity of composites against *S. mutans* ATCC 25175.

| Sample | CFU/mL After 4 h of Incubation | CFU/mL After 20 h of Incubation | RF % |
|---|---|---|---|
| Ti23Zr25Nb | $<1.0 \times 10^3$ | $3.2 \times 10^4$ | 78.67 |
| Ti23Zr25Nb-9BG | $<1.0 \times 10^3$ | $2.4 \times 10^4$ | 84 |
| Ti23Zr25Nb-9BG-Ag | $<1.0 \times 10^3$ | $3.1 \times 10^3$ | 97.93 |
| Ti23Zr25Nb-9BG-Cu | $<1.0 \times 10^3$ | $8.5 \times 10^3$ | 94.33 |
| Ti23Zr25Nb-9BG-Zn | $<1.0 \times 10^3$ | $3.5 \times 10^3$ | 97.67 |
| microcrystalline Ti (control) | $<1.0 \times 10^3$ | $2.0 \times 10^5$ | - |

CFU—colony-forming unit; RF—reduction factor.

## 4. Discussion

The crystal structure evaluation of the obtained materials clearly shows that the addition of 45S5 Bioglass limits the β stabilizing effect of niobium and zirconium. In its lower concentrations, only α and β phases peaks were visible. However, with higher concentrations of 45S5 Bioglass and elements such as silicon, calcium, sodium, and phosphorous, completely different phases than thos evident titanium-based solutions were present (Figure 2). It means that mechanical alloying leads to the homogenous distribution of 45S5 Bioglass components in the titanium matrix (Figure 7) but also destabilizes it during both milling and sintering. It could have been changed with the short milling of the proposed 45S5 Bioglass powders with the pre-milled Ti23Zr25Nb powders. This type of composite producing technique should lead to the mainly homogenous distribution of the amorphous 45S5 Bioglass phase in the beta-phase Ti23Zr25Nb alloy matrix. The main disadvantage of this method would be an aggregation of the 45S5 Bioglass and limitation of the fine-grained structure biocomposite formation. The proposed method of producing TiNbZr-based composites destabilized 45S5 Bioglass during the process but gave interesting mechanical, corrosion, and antibacterial results in terms of their biomedical applications.

With the higher content of 45S5 Bioglass, a significant reduction in the crystallite size could be observed. It was because of the limitation of the cold welding effect during milling and the intensification of plastic deformation and structure fracturing of the powders. It was visible both with much broader Ti (β) XRD peaks (Figure 5) and based on the Williamson–Hall linear plots. The size of the crystallite size can be limited even below 4 nm for Ti23Zr25Nb-6BG, Ti23Zr25Nb-9BG, Ti23Zr25Nb-9BG-Ag (Figure 3).

The limitation of Ti (β) with 45S5 Bioglass content was caused by its α-stabilizing effect, which was mainly caused by the major oxygen presence in its composition. This type of effect has also been observed in other β-type alloys as Ti-Mo alloys [54]. An increase in the Ti (β) lattice constant was observed in the Ti23Zr25Nb-3BG composite (Rietveld refinement results—Table 2). This content of 45S5 Bioglass did not allow the formation of phases other than titanium solutions: Ti (α) and Ti (β). That means that all of the 45S5 Bioglass elements were dissolved in the titanium solutions and mainly in the major Ti (β) phase leading to its extension.

Significant changes can be observed in the behavior of the alloy with the 6 wt.% content of 45S5 Bioglass. The precipitation of interstitial phases of $Ti_2ZrO$ and $Nb_5Si_3P$ were present because introduced elements could not be dissolved in the titanium solutions. Furthermore, the limitation of Zr content in the cubic Ti (β) phase led to the drastic decrease in its amount and contraction of its lattice constants (in contrast to the 3 wt.% content of 45S5 Bioglass).

The further increase in the 45S5 Bioglass content to 9 wt.% led to the progressing precipitation of the $Nb_5Si_3P$ and finally to a 65% content of Ti (β) in the produced composites. Furthermore, because of no $Ti_2ZrO$ participation, the Ti (β) lattice constant extended to the highest values of all the produced materials—3.3720 Å.

The addition of silver was proven to have some beta-stabilizing effect on the Ti23Zr25Nb-9BG composite. Despite the Zr limitation in the titanium solution because of the $Ti_2ZrO$ phase formation, the addition of Ag increased the amount of the Ti (β) phase. The drastic contraction of a unit cell and a decrease in the lattice constant to 3.3579 Å was also observed.

In the case of alloys with other antibacterial additives, a reduction in the Ti (β) phase content was visible despite the beta-stabilizing effect of Cu and Zn. The loss of niobium in $Nb_5Si_3P$ precipitates was too high and could not be replaced with both these elements, which tend to create eutectoid phase diagrams with titanium [55,56] and not isomorphous ones as does niobium [57].

In Ti23Zr25Nb-9BG-Ag and Ti23Zr25Nb-9BG-Zn, similar phenomenon as to those observed in Ti23Zr25Nb-6BG can be observed. The formation of $Ti_2ZrO$ limited the Zr soluted in the Ti (β) phase, which led to its contraction in contrast to Ti23Zr25Nb-9BG and Ti23Zr25Nb-9BG-Cu composites.

$Ti_2ZrO$ presence in the selected materials: Ti23Zr25Nb-6BG, Ti23Zr25Nb-9BG-Ag, and Ti23Zr25Nb-9BG-Zn might have been caused by the interfacial reaction between titanium and

zirconia formed from the oxygen being in the 45S5 Bioglass content during sintering. This type of reaction could lead to the formation of a hexagonal and oxygen-deficient $Ti_2ZrO$ phase [58].

The distribution of all elements in the titanium matrix was homogenous, which also corresponded to the homogenous distribution of all precipitates (Figure 7). The same type of distribution can also be observed in materials with the addition of Ag, Cu, and Zn (Figure 8) with none of this element being aggregated in produced composites but regularly dispersed in its volume and materials matrix. This type of composite is possible and easy to produce with mechanical alloying, which is absolutely the novel approach.

The Young modulus of all composites makes them suitable for biomedical applications. The most interesting could be the modulus of Ti23Zr25Nb-9BG, which is equal to 43.3 GPa and significantly lower than that of other β-type titanium alloys produced by members of our group [1–3]. However, there are still porous structures with significantly lower Young modulus' than those or Ti-based materials produced with the gel casting techniques, which could be one of the methods of reducing the modulus' of this type of material further [59]. The elastic modulus of the human bone can be measured as a wide range of values (from 4 to 30 GPa), depending on the testing parameters such as measuring direction and the exact type of the examined bone tissue [60,61]. With the nanoindentation using the Berkovich tip and the Oliver–Pharr indentation method, the Young modulus of bone was estimated as equal to 20 GPa and 10 GPa for dry and wet bone, respectively [62]. That means that the Ti23Zr25Nb-9BG Young modulus is still far from the modulus of bone but much closer than most titanium-based alloys. This composite allows us to limit the stress-shielding effect caused by the high elastic modulus of the implant and all the harmful effects of that phenomenon such as bone resorption, implant loosening, and implant failure [61,63]. The lower modulus of this composite could also be caused by its high porosity (around 20%), which should also lead to intensified osseointegration [64]. This limitation of the Young modulus goes in tandem with a decrease in the hardness of this composite. However, it is still beneficial in contrast to commercially pure titanium [1]. The mechanical properties of this composite are also coincidentally similar to the lattice constant of the T (β) phase, which is significantly the highest of all the tested materials (3.3720 Å). It might prove that in the Ti (β) phase, most bioglass components such as phosphorous and calcium, along with zirconium, which was proven in our previous research to have a high impact on the lattice constant, have dissolved. In contrast, for the hardest Ti23Zr25Nb-9BG-Zn, the hardness and lattice constant were equal to 387 HV0.3 and 3.3557 Å, respectively.

The surface free energy of all alloys was in the range of about 30–40 mN/m, contact angles were in the range of about 45–60° for diiodomethane, and about 60–80° for glycerol. That means that the wettability of all the composites was high, and all of them can be named as hydrophilic, with no significant differences between all of them (Table 4). Formation of the hydrophilic surface should not inhibit bone tissue growth on the produced materials, which could happen with the hydrophobic surface due to the high affinity to the broad types of proteins of this type of surface [65].

The corrosion current decreased, and the corrosion potential increased with the 45S5 Bioglass content (Figure 11). However, Ti23Zr25Nb-6BG and Ti23Zr25Nb-9BG alloys showed a drastic increase in the corrosion current around +0.5 V potential, which corresponds to the changes in the passive film. Both these composites should not be used in these ranges. Above +1 V there was once again the constant corrosion current and the composites should not corrode, being in the range of anodic protection. The Ti23Zr25Nb-3BG composite potentiodynamic curve was similar to that of Ti23Zr25Nb and pure titanium despite the higher corrosion current and the lower corrosion potential. The increase in the corrosion resistance of alloys containing 45S5 Bioglass could be caused by the significant changes in its passive film composition. This was observed by other researches on other types of alloys as magnesium alloys. The surface of the composite might be enriched with calcium and phosphate, leading to the formation of chemical compounds containing both of these elements [66,67].

The Ti23Zr25Nb-9BG-Ag composite had the lowest corrosion current of all the composites (Figure 12). It is well known and researched that the addition of a noble metal such as silver to the alloy

can lead to that because of the increase in the passive film stability and the easiness of passivation [68]. The addition of copper tends to increase the corrosion current to the highest corrosion current among all the composites. It could be caused by the copper enrichment of the passive film and its impoverishment with the other alloy components such as Ti, Zr, Nb, Ca, and P. Other research groups previously observed this type of change in the passivation behavior of titanium-based alloys [69]. The same type of mechanism may provide the increase in corrosion current of the Ti23Zr25Nb-9BG-Cu composite in contrast to the Ti23Zr25Nb-9BG composite. The curve of Ti23Zr25Nb-9BG-Zn looks most familiar to Ti23Zr25Nb-9BG, with the same type of passivation observed in its anodic part. Furthermore, Ti23ZrNb-9BG-Zn and Ti23Zr25Nb-9BG corrosion currents and potentials were in the same order of magnitude.

In the present research, the ability of *S. mutans* ATCC 25175 to form biofilm on the ultrafine-grained Ti23Zr25Nb alloy and subjected to the different types of chemical modifications, including 45S5 Bioglass, Ag, Cu or Zn was evaluated. *S. mutans* is a Gram-positive bacteria commonly found in the human oral cavity and is the main contributor to tooth decay. The microbe was first described by J. Kilian Clarke in 1924 [70].

The obtained results indicate that the tested *S. mutans* reference strain can adhere to the Ti23Zr25Nb-9BG composites produced by the powder metallurgical method. Additionally, the type of chemical modification (Ag, Cu, Zn) influences the ability of *S. mutans* to form biofilm on the tested biomaterial.

According to published research, when a Ti23Zr25Nb-9BG-Ag composite stays immersed in body fluid, silver could react with the environment and release ionized Ag into the surrounding area [71]. A release of the silver biocide at a concentration level (0.1 ppb) is capable of rendering the antibacterial efficacy [71,72].

The mechanism for bacterial toxicity of the tested Ti23Zr25Nb-9BG-Ag alloy may include the free metal ion toxicity arising from the dissolution of metals from the surface of the silver particles (e.g., Ag+ from Ag) [72,73] or the oxidative stress via the generation of reactive oxygen species (ROS) on crystal surfaces of some nanoparticles [74].

Copper and copper alloys were able to decrease the counts of bacteria, including pathogens such as methicillin-resistant *Staphylococcus aureus* (MRSA), by seven to eight logs within hours [75]. The mechanism for bacterial toxicity of the tested Ti23Zr25Nb-9BG-Cu alloy was the same as in the case of Ag containing biomaterials.

Zinc is an environmentally friendly material and has little toxicity [76]. It is naturally present in dental plaque and saliva. At higher concentrations Zinc is toxic. For many years, Zn was incorporated into many dental materials due to the ability of zinc ions to inhibit the growth of cariogenic bacteria. Recently, it has been shown that zinc oxide nanoparticles (ZnO-NPs) have an antibacterial property [77]. Our findings allowed us to assess the effectiveness of zinc against *S. mutans*.

Titanium remains mostly neutral and is certainly not poisonous for the human body environment, which can tolerate this metal in large doses. Therefore, the composites based on the Ti containing 45S5 Bioglass and Ag (or Cu, Zn) additions have the potential to be used in dentistry with infection control. Generally, the Ti-base alloy containing niobium, zirconium (TiZrNb) is beneficial in dental and other medical device applications. Even though zirconium and other elements in Ti-based alloys for dental and medical uses are nontoxic, there are still ongoing studies to ensure that the materials themselves don't have adverse side effects over the long term [78]. Cossellu et al. found that there might be a link between zirconium implants and some health problems, such as inflammation and skeletal and connective tissue disorders [78]. Ultrafine-grained Ti23Zr25Nb-9BG-based composites possess unique mechanical properties and are thus considered to represent the next generation of biomaterials. Additionally, the addition of the Ag, Cu or Zn to Ti23Zr25Nb-9BG composites significantly lowered the adhesion of *S. mutans*, suggesting that these composites had antibacterial activity.

## 5. Conclusions

The main goal of these studies was to produce biocomposites based on the Ti23Zr25Nb alloy previously tested by our group [1]. The composites were formed by the addition of 45S5 Bioglass and three different antibacterial additives: silver, copper, and zinc. All experiments conducted on these novel materials allow us to withdraw some conclusions:

− the content of Ti (β) increases with milling time and 45S5 Bioglass decreases it,
− Ti23Zr25Nb-3BG and Ti23Zr25Nb-6BG need longer milling time (16 h) than Ti23Zr25Nb (10 h) to produce fully single-phase β-type powders,
− 45S5 Bioglass limits the content of Ti (β) as a beta non-stabilizer in the bulk samples,
− Cu and Zn decrease the content of Ti (β) despite their β–stabilizing properties in the bulk samples,
− the corrosion, mechanical properties, and wettability of the produced materials are beneficial in contrast to the Ti23Zr25Nb alloy,
− Ti23Zr25Nb-9BG-Ag, Ti23Zr25Nb-9BG-Cu, Ti23Zr25Nb-9BG-Zn are proved to have high antibacterial activity against *S. mutans* being present in the human oral cavity,
− produced materials based on the performed experiments are highly recommended for biomedical use (especially as dental implants) because of their high corrosion resistance, low Young modulus, and good antibacterial activity.

**Author Contributions:** Conceptualization, M.M., M.J.; Formal analysis, M.M.; Funding acquisition, M.J.; Investigation, M.M., P.P., M.R., M.G. and M.U.J.; Supervision, M.J.; Writing—original draft, M.M.; Writing—review and editing, M.M., M.J., P.P., M.R., M.G. and M.U.J. All authors have read and agreed to the published version of the manuscript.

**Funding:** The work has been financed by National Science Centre Poland (under decision no. DEC-2017/25/B/ST8/02494).

**Conflicts of Interest:** The authors declare no conflict of interest.

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
