# Peer review of "The Effect of 45S5 Bioglass and Ag, Cu, or Zn Addition on the Crystal Structure, Properties, and Antibacterial Effect of Bulk Ti23Zr25Nb Biocomposites"

_metals, doi:10.3390/met10091115_

Round 1
Reviewer 1 Report
Interesting paper about new materials for dental applications with a fairly classic study.
A better granulometry of the powders could be more efficient.
Author Response
Review 1
Dear Reviewer,
We would like to appreciate you allowing us to resubmit our manuscript: The effect of 45S5 Bioglass and Ag, Cu, or Zn addition on the crystal structure, properties, and antibacterial effect of bulk Ti23Zr25Nb biocomposites to Metals. We are pleased with your comments and objective feedback considering the draft of our article. Our responses to the comments were provided below:
Interesting paper about new materials for dental applications with a fairly classic study.
A better granulometry of the powders could be more efficient.
We agree with this comment and a too general statement of the particle size. Unfortunately, the best possible granulometry analysis could be done with the use of the proper image processing software or different methods than microscopic one. Unfortunately, none of them are available by the authors at the moment. However, the authors would consider adding more accurate granulometry analysis in their future studies. In that article more clear statement than the previous one was added:
Page 7 - “Most of them vary from a not strongly agglomerated particles in size less than 10 microns to the highly agglomerated particles in the size of the hundreds of microns with the biggest particles in the size of approximately 500 μm. The variation of the particle size is much higher than in the non-modified alloys [1].”
We would like to thank you once again for your suggestions for improving our manuscript.
Yours faithfully,
M.Jurczyk

Reviewer 2 Report
The manuscript deals with a very promising biocompatible material suitable, for example, for the production of dental implants. A lot of useful work has been done in the manuscript and in principle it could be accepted for publication in this state. My comments are recommendations for the following paper, but if the authors want, they can take them into account already in this paper.
1. Sintering is obviously not enough to consolidate the powder well. I would definitely not want a dental implant made of such a material. I therefore recommend using HIP or hot rolling or hot forging for its consolidation.
2. After consolidation, much more detailed mechanical testing is required, hardness measurement is not enough.
3. There is a lot of oxygen in the system, which must be present in the form of oxides after consolidation. The oxides are apparently not detected by XRD due to their small size. It is therefore necessary to detect these oxides by SEM or TEM to get an idea of their size and homogeneity. This should provide feedback for the required mechanical alloying time ensuring a homogeneous dispersion of oxides.
Author Response
Review 2
Dear Reviewer,
We would like to appreciate you allowing us to resubmit our manuscript: The effect of 45S5 Bioglass and Ag, Cu, or Zn addition on the crystal structure, properties, and antibacterial effect of bulk Ti23Zr25Nb biocomposites to Metals. We are pleased with your comments and objective feedback considering the draft of our article. Our responses to the comments were provided below:
Sintering is obviously not enough to consolidate the powder well. I would definitely not want a dental implant made of such a material. I therefore recommend using HIP or hot rolling or hot forging for its consolidation.
We agree with this comment. Hot isostatic pressing is the consolidation method that can significantly improve the quality of materials produced with powder metallurgy approach. We already proved that in one of our previous works (Marczewski, M.; Miklaszewski, A.; Jurczyk, M. Structure evolution analysis in ultrafine-grained Zr and Nb-based beta titanium alloys. J. Alloys Compd. 2018, 765, 459–469.). The HIP has allowed us to decrease the porosities in our materials from 20-30% to less than 2%. We surely consider your comment, and we will try to enhance our materials further by using HIP in our future work.
After consolidation, much more detailed mechanical testing is required, hardness measurement is not enough.
Thank you for that suggestion. We agree that hardness measurement is not enough for mechanical testing of biomedical materials. In our work, we decided to test Ti23Zr25Nb-BG composites additionally with nanoindenter to estimate Young modulus of these materials, which together gives really interesting pieces of information about them. However, our group has already published papers with compression testing results in the case of other titanium-based alloys with beta-stabilizers (Ta) and their foams. Results could be read in the article: G. Adamek, M. Kozlowski, M.U. Jurczyk, P. Wirstlein, J. Zurawski, J. Jakubowicz. Formation and properties of biomedical Ti-Ta foams prepared from nanoprecursors by thermal dealloying process, Materials (Basel). 12 (2019). We would try to do the same experiments for our composites.
- There is a lot of oxygen in the system, which must be present in the form of oxides after consolidation. The oxides are apparently not detected by XRD due to their small size. It is therefore necessary to detect these oxides by SEM or TEM to get an idea of their size and homogeneity. This should provide feedback for the required mechanical alloying time ensuring a homogeneous dispersion of oxides.
Authors go along with that comment. TEM should be treated on the produced materials. It would present the distribution of oxides and other phases in the composites. Authors will take that into account and try to find the coincidence between the time of milling, crystal structure, and the distribution of the particular phases in one of their future work considering these materials.
We would like to thank you once again for your suggestions for improving our manuscript.
Yours faithfully,
M.Jurczyk

Reviewer 3 Report
Unfortunately, the referee could not consider this paper to be a manuscript. Rather, I felt it was a technical report. For this reason, I had no choice but to reject it.
From the Abstract, we cannot read the scientific idea why this material was modified with additive elements such as Ag. Moreover, the introduction does not make clear what the research aims to achieve. For this reason, many research cases are referred to, but it is not possible to read what the technical problems are to achieve the purpose. The ambiguity of this purpose makes it difficult to read the goals of this study.
This paper contains a lot of valuable experimental data, but I am not able to correctly judge the validity of the experimental method and the experimental results because the achievement goal of the research is unknown. Therefore, I strongly recommend that the goals of this research be clarified by correctly extracting the technical issues from the past research, and then submitting again.
Author Response
Review 3
Dear Reviewer,
We would like to appreciate your review considering the draft of our article The effect of 45S5 Bioglass and Ag, Cu, or Zn addition on the crystal structure, properties, and antibacterial effect of bulk Ti23Zr25Nb biocomposites to Metals despite not so positive response we could expect. We are pleased with your comments and objective feedback considering the draft of our article. Our responses to the comments were provided below:
From the Abstract, we cannot read the scientific idea why this material was modified with additive elements such as Ag. Moreover, the introduction does not make clear what the research aims to achieve. For this reason, many research cases are referred to, but it is not possible to read what the technical problems are to achieve the purpose. The ambiguity of this purpose makes it difficult to read the goals of this study.
The authors appreciate that comment but not fully agree with it. The last paragraph of the introduction part presents the purpose and aims of the studies:
“The reason for the modification with 45S5 Bioglass was to improve the properties which consider lowering Young modulus and improving the corrosion resistance of the non-modified Ti23Zr25Nb alloy. Corrosion resistance has been revealed as worse than that of microcrystalline titanium in our previous work [47]. This type of bioglass is providing good cell growth and differentiation of osteoblasts due to the low phosphate content and the widest clinical application among all other bioglasses [50,51]. The addition of well-know elements with antibacterial properties as Ag, Cu, and Zn was to enhance the antibacterial response of the material, which should make the produced material interesting in use in the biomedical field. The influence of microstructure and chemical composition of Ti23Zr25Nb and Ti23Zr25Nb-BG-Ag (or Cu, Zn) composites on the crystal structure, microstructure, mechanical properties, corrosion behavior, surface wettability, and antibacterial activity against reference strain of Streptococcus mutans were investigated in detail.”
Moreover, additional advantages of producing these types of composites were presented by citing numerous articles in the field with some of them published by the members of our group based on our previous studies:
Jurczyk, K.; Miklaszewski, A.; Jurczyk, M.U.; Jurczyk, M. Development of β type Ti23Mo-45S5 bioglass nanocomposites for dental applications. Materials (Basel). 2015, 8, 8032–8046
Jurczyk, K.; Kubicka, M.M.; Ratajczak, M.; Jurczyk, M.U.; Niespodziana, K.; Nowak, D.M.; Gajecka, M.; Jurczyk, M. Antibacterial activity of nanostructured Ti-45S5 bioglass-Ag composite against Streptococcus mutans and Staphylococcus aureus. Trans. Nonferrous Met. Soc. China (English Ed. 2016, 26, 118–125.
This paper contains a lot of valuable experimental data, but I am not able to correctly judge the validity of the experimental method and the experimental results because the achievement goal of the research is unknown. Therefore, I strongly recommend that the goals of this research be clarified by correctly extracting the technical issues from the past research, and then submitting again.
The main research goals of these studies were presented in the introduction (especially in its last part) with the technical issues considering the Ti23Zr25Nb alloys. It was written why using 45S5 Bioglass and antibacterials additives could lead to the improvement to our Ti23Zr25Nb alloy which is already interesting among other Ti-based alloys:
Marczewski, M.; Miklaszewski, A.; Jurczyk, M. Structure evolution analysis in ultrafine-grained Zr and Nb-based beta titanium alloys. J. Alloys Compd. 2018, 765, 459–469.
Marczewski, M.; Miklaszewski, A.; Maeder, X.; Jurczyk, M. Crystal Structure Evolution, Microstructure Formation, and Properties of Mechanically Alloyed Ultrafine-Grained Ti-Zr-Nb Alloys at 36 ≤ Ti ≤ 70 (at. %). Materials (Basel). 2020, 13.
The sense of this modification was proved with the Results and then the Discussion part. Summarizing of all achievements was done in the Conclusion part:
Pages 21-22 - “The main goal of these studies was to produce biocomposites based on the Ti23Zr25Nb alloy previously tested by our group [1]. The composites were formed by the addition of 45S5 Bioglass and 3 different antibacterial additives: silver, copper, and zinc. All experiments conducted on this novel materials allow us to withdraw some conclusions:
- the content of Ti(β) increases with milling time and 45S5 Bioglass decreases it,
- Ti23Zr25Nb-3BG and Ti23Zr25Nb-6BG need longer milling time (16 h) than Ti23Zr25Nb (10 h) to produce fully single-phase β-type powders,
- 45S5 Bioglass limits the content of Ti(β) as a beta non-stabilizer in the bulk samples,
- Cu and Zn decreases the content of Ti(β) despite their β–stabilizing properties in the bulk samples,
- the corrosion, mechanical properties, and wettability of produced materials are beneficial in contrast to the Ti23Zr25Nb alloy,
- Ti23Zr25Nb-9BG-Ag, Ti23Zr25Nb-9BG-Cu, Ti23Zr25Nb-9BG-Zn are proved to have high antibacterial activity against mutans being present in the human oral cavity,
- produced materials based on performed experiments are highly recommended for biomedical use (especially as dental implants) cause of their high corrosion resistance, low Young modulus, and good antibacterial activity.”
Moreover, because of the 45S5 Bioglass improving the bioactivity of the materials, we want to do the biological tests of these composites in contrast to the non-modified TiZrNb alloys. The cytocompatibility tests (MTS assay) are now in progress. The obtained final results will be deeply analyzed and published in one of our future articles.
We would like to thank you once again for your suggestions for improving our manuscript.
Yours faithfully,
M.Jurczyk

Reviewer 4 Report
the introduction provides sufficient background, methods adequately described, the results clearly presented, and the conclusions supported by the results. therefore, it should be accepted for publication in present form.